# Broadening the Horizons of RNA Delivery Strategies in Cancer Therapy

**DOI:** 10.3390/bioengineering9100576

**Published:** 2022-10-19

**Authors:** Shuaiying Wu, Chao Liu, Shuang Bai, Zhixiang Lu, Gang Liu

**Affiliations:** 1State Key Laboratory of Molecular Vaccinology and Molecular Diagnostics & Center for Molecular Imaging and Translational Medicine, School of Public Health, Xiamen University, Xiamen 361102, China; 2State Key Laboratory of Cellular Stress Biology, Innovation Center for Cell Biology, School of Life Sciences, Xiamen University, Xiamen 361102, China

**Keywords:** RNA delivery, delivery carrier, cancer therapy, clinical practice

## Abstract

RNA-based therapy is a promising and innovative strategy for cancer treatment. However, poor stability, immunogenicity, low cellular uptake rate, and difficulty in endosomal escape are considered the major obstacles in the cancer therapy process, severely limiting the development of clinical translation and application. For efficient and safe transport of RNA into cancer cells, it usually needs to be packaged in appropriate carriers so that it can be taken up by the target cells and then be released to the specific location to perform its function. In this review, we will focus on up-to-date insights of the RNA-based delivery carrier and comprehensively describe its application in cancer therapy. We briefly discuss delivery obstacles in RNA-mediated cancer therapy and summarize the advantages and disadvantages of different carriers (cationic polymers, inorganic nanoparticles, lipids, etc.). In addition, we further summarize and discuss the current RNA therapeutic strategies approved for clinical use. A comprehensive overview of various carriers and emerging delivery strategies for RNA delivery, as well as the current status of clinical applications and practice of RNA medicines are classified and integrated to inspire fresh ideas and breakthroughs.

## 1. Introduction

Cancer has always been a major public health problem to be solved, and despite significant advances in existing therapies in recent years, its treatment remains a formidable challenge [1]. Drug therapy is the most conventional strategy; however, nonspecificity and low bioavailability are the core factors leading to the poor therapeutic efficacy [2]. The occurrence and development of the tumor are closely related to gene mutation, so gene therapy has been widely concerned in the field of anti-tumor [3,4,5]. Therapeutic genes can be transported into target cells through appropriate carriers to precisely up-regulate/down-regulate the expression of specific tumor-related genes, inhibiting the occurrence and development of tumors, thus achieving efficient cancer treatment [6,7]. Actually, gene therapy has already been widely carried out in scientific research and clinical trials. Noteworthily, the U.S. Food and Drug Administration (FDA) recently approved a gene therapy drug onpattro for the treatment of polyneuropathy, indicating that the technology has fine clinical prospects. Ribonucleic acid (ribonucleic acid, RNA) is the genetic information carrier of biological cells, and is critical for gene expression. Many different types of RNA have been shown to be dysregulated in tumors [8]. RNA therapeutics have the advantage of being more convenient than protein-based drugs [9,10,11]. In 1958, Crick’s central law mentioned that RNA plays a crucial role in the transmission of genetic information for the first time [12]. In subsequent studies, RNA was found to be capable of complementary base pairing to form a double-stranded structure similar to DNA, which laid a foundation for the discovery of mRNA-interfering complementary RNA (micRNA) and the emergence of RNA interference drugs [13]. Complementary pairing of RNA bases was first used to treat diseases in 1978 [14]. The RNA interference strategy was first described in 1998 in the study of Andrew et al. [15]. Two decades later, the first siRNA related drug prescribed to treat hereditary thyroxine transporter-associated amyloidosis was approved. In addition to the therapeutic use of RNA interference, messenger RNA (mRNA) is also used in the prevention and treatment of diseases. An mRNA vaccine for cancer treatment was first designed in 1995 [16]. In 2008, the results of the first clinical trials of mRNA vaccines were reported [17]. Most recently, the first mRNA vaccine against SARS-CoV-2 was approved in 2020 [18].

The recent explosion of RNA research has provided new methods for the synthesis and delivery of RNA [19]. Nonetheless, naked RNA therapy has presented prominent drawbacks, such as (i) short half-life due to nuclease degradation and clearance by reticuloendothelial cells; (ii) negative charge and molecular weight cause low cellular uptake efficiency; (ii) a series of obstacles such as targeting tumor tissue, endocytosis, and endosomal escape; (iv) in addition, there is a risk that RNA may be off-target, causing mutations in normal genes to induce more serious accidents [20,21]. Improving bioavailability and protecting stability are the key factors to achieving RNA delivery in vitro and in vivo and ultimately improve the efficacy of tumor therapy.

To overcome these obstacles and achieve the ideal therapeutic purpose, gene carrier has become the focus of gene therapy research. Different carriers have been developed including cationic polymers [22,23], liposomes [24], exosomes [25], inorganic nanomaterials [26], DNA structures [27], protein structures [28], etc. These vectors can provide better protection against RNA degradation in the blood circulation and can be designed to neutralize RNA’s negative charge to allow for more efficient endocytosis, as well as target modification to enable active targeting and more accurate action on target genes. Various gene carriers have been reported that need to be comprehensively and systematically reviewed in order to better grasp the research trends, understand their potential value, and inspire researchers to develop new delivery systems. Therefore, we hereby systematically summarize the up-to-date delivery strategies of RNA delivery (Figure 1), focusing on the advantages and disadvantages of different carrier types, as well as the clinical translation, challenges, and application prospects of RNA in cancer treatment. We expect that such a comprehensive review will provide important information about gene-mediated carriers and progress of gene therapy for interested researchers in this field, and meanwhile, we hope it will inspire new ideas for the design and development of various delivery systems in the future.

## 2. The Challenge of RNA Delivery

### 2.1. Strategies for Using RNA Interference

RNA interference (RNAi) technology utilizes double-stranded RNA (dsRNA) to efficiently and specifically degrade intracellular homologous messenger RNA, thereby blocking specific gene expression and resulting in a target gene deletion phenotype. Common effector molecules mainly include small interfering RNAs (siRNAs), micro RNAs (miRNAs), and short hairpin RNAs (shRNAs) [29,30]. Given its unique RNAi-triggering activity, siRNA shows great potential in RNA-mediated therapy. siRNA is a 19–23 base pair double-stranded RNA with two nucleotides at the 5′-phosphorylated and unphosphorylated 3′-terminal [31]. RNA nucleases bind to double-stranded RNA and enzymatically chop it into siRNA segments of 21–25 base pairs in size. Subsequently, siRNA binds to the polymerase complex (RISC) and performs the effector function of RNA interference through the corresponding messenger RNA (mRNA) site. The enzyme degrades the mRNA, resulting in the termination of transcriptional gene expression, and ultimately leads to the reduction of gene expression [32,33]. MicroRNAs (miRNAs) are a class of non-coding single-stranded RNA molecules, which are encoded by endogenous genes and are about 22 nucleotides in length. It is closely related to siRNA, and these non-coding RNA molecules (ncRNAs) are involved in the regulation of gene expression, but its mechanism is different from siRNA-mediated mRNA degradation. RISC typically hybridizes with partial complementary binding sites on three untranslated regions of the target mRNA, or binds to the target mRNA and facilitates its cleavage [34,35].

siRNA faces both internal and external barriers. The internal drawback is the short lifespan of synthetic siRNAs, and their ability to regulate gene expression is impaired. Two strategies have emerged to alleviate this dilemma: (i) the introduction of chemical modifications to the backbone of oligonucleotides can extend their half-life and (ii) in vivo processing to short hairpin-like RNA transcripts (shRNA) that express siRNA [36,37,38]. The shRNA consists of two short reverse repeats. shRNAs cloned into shRNA expression vectors consist of two short reverse repeats separated by a stem-loop sequence, which form a hairpin structure and are controlled by pol III promoters. Subsequently, 5-6 T bases were attached as transcriptional terminators of RNA polymerase III. siRNA can be cloned into plasmid vectors, and when entered in vivo, the hairpin sequence is expressed to form a “double-stranded RNA” (dsRNA) that is processed by RNAi channels. Despite some achievements, siRNA-mediated gene silencing therapy still faces many challenges in achieving its desired goals: (i) siRNA is easily degraded: the rich ribonuclease in serum will degrade the naked siRNA and block siRNA from further functioning. Unfortunately, the half-life of siRNA in the blood is less than 10 min. (ii) Delivery of siRNA is challenging: the size and hydrophilicity of siRNA prevent it from crossing the cell membrane. (iii) Insufficient targeting accuracy: the blood vessels in tumor tissue are leaky and twisted, irregular in shape, too sparse in some areas, and too dense in others. These malformations lead to abnormal blood flow and a tendency for blood to leak out of the blood vessels. (iv) Off-target effects: the introduction of siRNA may lead to abnormal expression of other genes besides the target gene. Such an off-target effect is an undesirable side effect of the specific knockdown of the target gene. There are two main possible off-target factors, such as (a) the siRNA seed region pairs with the unexpected gene, forming an incomplete complementary pair that continues its gene silencing effect; (b) the level of RNAi in vivo is saturated when external siRNA is transported into cells, they compete with endogenous miRNAs and are therefore disturbed [36,37,38,39,40,41].

### 2.2. Strategies for Using mRNA

Messenger RNAs (mRNAs) play a crucial role in the central tenets of molecular biology developed by Francis Crick. mRNA is defined as the medium through which information flows from DNA into cells [42,43]. As the structure and function of mRNA have been gradually revealed in the past few decades, mRNA has been considered the most attractive alternative to protein drugs, so the application value and potential of mRNA have been increasingly concerned [44]. Nowadays, given the advancement of mRNA chemical modification and nucleic acid delivery carrier, it has shown important application value in tumor immunotherapy [45,46,47]. Although mRNA drugs have great application prospects, many current challenges must be fully addressed before mRNA drugs can be successfully used in clinical practice [48]. First, mRNA is very unstable in the bloodstream and is easily degraded by abundant ribonuclease enzymes [49,50]. To ensure effective mRNA function, in vivo stability must be greatly improved. Second, the injection of the mRNA triggers the human immune system to reject it and allow it to be quickly cleared [51]. The negatively charged property of mRNA makes it unlikely to actively cross anionic membranes with a hydrophobic inner layer without carrier encapsulation. Moreover, compared with siRNA, mRNA has a larger molecular weight. There are even greater obstacles in the process of penetrating the cell membrane. After entering the target cell, the lysosomal escape ability of mRNA is another challenge. These ongoing challenges prevent mRNA from being widely used in biomedicine [47]. Therefore, many efforts have been made to address these issues recently.

Therefore, to overcome a series of obstacles such as nuclease degradation, targeted tumor tissue, endocytosis, and endosomal escape, it is particularly important to realize in vitro and in vivo delivery of gene vector-mediated therapeutic nucleic acid to maximize the therapeutic effect of cancer. The design and development of efficient gene vectors is the focus of research in the field of gene therapy. The reported RNA delivery carrier will be summarized and introduced in the following sections.

## 3. Classification of RNA Drug Carrier

### 3.1. Cationic Polymers

Polymers have been identified as a potential delivery method for siRNA therapy [52,53]. Many groups use the high negative charge of nucleic acids to pass through RNA statically recombined with cationic substances. Positively charged polymers such as poly (L-lysine) (PLL), polyethyleneimine (PEI), polyamide (PAMAM), and poly (β-amino ester) (PBAE) can lead to the condensation of RNA by electrostatic interaction with negatively charged phosphate groups on the main chain of RNA and provide stability to prevent nuclease degradation [54]. In addition, cationic polymers have considerable proton buffering capacity, which facilitates the escape of RNA-bound endosomes. Under acidic conditions, negatively charged RNA is protonated in endosomal vesicles. This proton sponge property results in an influx of negative ions into the vesicle, leading to an increase in vesicle osmotic pressure, which eventually leads to membrane rupture and the release of RNA into the cytoplasm, successfully achieving endosomal escape [55,56]. Among cationic polymers, PEI as a commonly used synthetic polymer gene carrier has been proven to be an effective transfection agent for delivering RNA to a variety of normal, tumor, or stem cells to facilitate in vitro and in vivo RNA delivery and gene therapy [57,58,59,60]. Polyethyleneimine, as a well-known polymer, mainly has two forms: linear chain and branched chain. Branched structures are created by acid-catalyzed polymerization of aziridine monomers. Similarly, at lower temperatures, a similar approach can also obtain PEIs with linear structures. Ethyleneimine repeat unit makes these polymers highly water-soluble. High cationic charge density is one of the most important properties of these molecules, and one potentially portable amino nitrogen consists of three atoms. At physiological pH, only 20% or 16.7% of the amino nitrogen were protonated from the relationship between free PEI protonation and pH [61]. The combination of RNA and PEI had a weak effect on the protonation distribution of PEI, and only two or three nitrogen would be protonated at physiological pH. This shows that PEI has a high buffering capacity with a very wide pH range compared to other polymers [62,63].

Despite the high transfection efficiency of PEI, its cytotoxicity and non-biodegradability are unavoidable. In this case, improving the biocompatibility of PEI is a requisite. Wen and colleagues constructed a pH-responsive carrier polymer for delivery of PD-L1-siRNA using biocompatible hyaluronic acid (HA) combined with polyethyleneimine (PEI) and matrix metalloproteinase-2 (MMP-2). It was confirmed that the synthesized polymer had lower cytotoxicity and faster cell uptake capacity, improved the penetration of PD-L1-siRNA into lung tumor spheres, and effectively down-regulated the expression of PD-L1 in H1975 cells [64]. Luo et al. constructed HAP-PEI nanoparticles consisting of PEI-modified hydroxyapatite (HAP). The siRNA of KRAS loads genes (siKRAS) onto the surface of HAP-PEI through electrostatic interaction between siRNA and PEI to design functional HAP-PEI nanoparticles (HAP-PEI/siKRAS) for anti-pancreatic cancer therapy. The results show that KRAS expression genes were effectively knocked down, and KRAS protein expression was down-regulated in vitro. Otherwise, HAP-PEI shows remarkable cellular safety in normal pancreatic HPDE6-C7 cells [65]. PEI can be further modified with fluoroalkyl chains to give its surface hydrophobic and oleophobic properties. In Yuan’s study, according to the fluorination of anhydride reaction, the low molecular weight of 1.8kDa PEI was used to prepare PEIF. The results revealed that PEIF/siRNA complexes have suitable and stable particle sizes and the potential to compress nucleic acids at extremely low ratios, and have high efficiency in silencing specific genes, low cytotoxicity, and good tumor suppressive effects [66] (Figure 2A).

To alter biodistribution and further improve the stability, efficacy, and biocompatibility of NPs, PEI-based systems have been modified with liposomes to produce lipo-polymers that combine the properties of both systems [70]. The research by Zhupanyn et al. first explore the combination of PEI-based nanoparticles with native ECVs from different cell lines to deliver small RNAs. The loading efficiency and storage stability of PEI/siRNA complexes modified with ECV were demonstrated. In vivo treatment studies in mice with prostate tumors showed that after treatment with PEI/siRNA-ECVs, tumor growth was significantly inhibited and target genes were significantly knocked down [68] (Figure 2C). In Figure 2B, a novel hypoxia-sensitive nanoparticles system (PEG-AZo-PEI-Dope (PAPD)) was prepared by Joshi et al., which is mainly composed of PEG 2000, azobenzene (Azo), polyethyleneimine (PEI), and 1, 2-dioleyl-Sn-glycero-3-phosphate ethanolamine (DOPE) units. Through electrostatic interaction, PAPD nanoparticles form complexes with siRNA and achieve high efficiency in encapsulating many siRNA molecules within each particle [67]. It can be seen that PEI is highly efficient as a delivery carrier but has no targeting, and at the same time, the excess positive charge is accompanied by obvious cytotoxicity, which limits its further clinical application.

Chitosan is one of the main cationic polymers and the second most abundant polysaccharide in nature [71].Chitosan is also a natural biopolymer, which is composed of β-(1→4)-2-acetamino-D-glucose and β-(1→4)-2-amino-D-glucose units [72]. One primary amine and two free hydroxyl groups constitute each chitosan monomer, and its unit formula is C_6_H_11_O_4_N [73]. Its superior biocompatibility and biodegradability have been well studied and widely used in biomedicine [74,75]. In addition to good biodegradability, biosafety, low immunogenicity, and low toxicity, chitosan can effectively coagulate negatively charged nucleic acid and prevent nucleic acid from being degraded in nuclease or serum by combining its rich amine group with lotus through classical interaction force [76]. In 1995, chitosan was first reported as a non-viral gene delivery system for plasmid transfection [77]. In 1998, the application of chitosan in vivo and its potential for delivering nucleic acids were confirmed. Subsequently, several studies have been conducted on the potential use of chitosan and its derivatives in DNA delivery [78]. In 2006, chitosan was first used for the delivery of siRNA clinically [79]. So far, more and more studies have confirmed the feasibility and potential of chitosan as a nucleic acid delivery carrier [80,81].

However, the poor solubility of chitosan under neutral and alkaline conditions limits its applicability and application range. It is an effective strategy to improve the solubility of chitosan by chemical modification of its quaternized derivative N, N, N-trimethyl chitosan (TMC) to improve the physicochemical properties [82]. Masjedi et al. combined TMC with hyaluronic acid (HA) to form nanoparticles and then adsorb siRNA through electrostatic interaction force. The results confirmed that the prepared HA-TMC NPs possessed outstanding physicochemical performances, exhibited high loading capacity for siRNA, and significantly improved cellular uptake efficiency by specifically binding to CD44 on cancer cells (Figure 2D) [26]. Similarly, Liang et al. constructed hyaluronic dialdehyde (HAD) in an ethanol–water mixed system for CD44-targeted siRNA delivery by covalently binding to chitosan nanoparticles (CS-HAD NPs). These results indicated that CS-HAD NPs can not only target the CD44 receptor but also deliver therapeutic siRNA to T24 bladder cancer cells through a ligand receptor-mediated targeting mechanism to achieve significant treatment [83]. In addition, carboxymethyl chitosan (CMC), formed by carboxymethylation of hydroxyl or amino groups on chitosan, is a water-soluble chitosan derivative. The reversible protonation and deprotonation properties of amines and hydroxyl groups contribute to the two characteristics of CMCs, that is, their ability to act as a hydrogen ion sponge, and their surface charge is pH-dependent [84]. Peng et al. utilized the crosslinker SPDP to react with the amino group of CMC and the sulfhydryl group of DOX-SH to form a CMC-DOX nanoplatform, which contains disulfide bonds (Figure 2E). With the ratio of 1, low polyethylene imine (OEI) was added to form an exfoliate shield shell to coagulate siRNA, and the final encapsulation efficiency was always higher than 89% [69].

### 3.2. Exosomes

Extracellular vesicles (EVs) are small membrane vesicles released by cells into the extracellular matrix, which have the characteristics of participating in the exchange of information between cells. Compared with commonly used lipid nanoparticles (LNPs) delivery systems, exosomes have their own unique advantages: (i) It is stable and showing a difficult clearance by macrophages, thus providing better drug protection during drug delivery [85]; (ii) relatively low cytotoxicity and immunogenicity, and high biosafety [86]; (iii) capable of transporting both hydrophobic and hydrophilic molecules, multivalent display of cell surface groups with efficient homing to tumor sites [87]; (iv) homologous exosomes can recognize each other, send out a “don’t eat me” signal, own escape the mononuclear phagocyte system, and effectively deliver drugs [88]; (v) the presence of membrane proteins such as tetraspanins and fibronectin confers high cellular uptake efficiency of exosomes and allows easy modification according to target cells [89]. miRNA, one of the non-coding single-stranded RNA molecules, is often protected and delivered by exosomes to resolve its instability. In the study of Kim et al., they used exosomes of HEK293T cells as the carrier of let7c-5p-miRNA to treat breast cancer, and the results clearly showed that let7c-5p can effectively regulate the growth and migration of cancer cells [90]. Similarly, Forterre et al. used this approach for mRNA delivery in the treatment of breast cancer. Human embryonic kidney 293 (HEK293) cell-generated extracellular vesicles for HChrR6 mRNA delivery for the treatment of HER2^+^ breast cancer. The final results showed that ML39 SCFV and Tretazicar’s systemic IVT mRNA-loaded EVs at lower EV doses than previously caused almost complete growth arrest of human HER2^+^ breast cancer xenografts in non-thyroid mice [91]. Kaban designed a novel strategy based on exosomes and targeted Bcl-2 using transgenic natural killer (NK) cells. Exosomes (NKExos) encapsulated Bcl-2 siRNA (siBCL-2) transduced by lentivirus for expression and further evaluated in ER+ breast cancer. Large amounts of Bcl-2 siRNA were produced by transfected NK92MI cells but did not significantly affect NK cell viability or effector function (Figure 3A). siBCL-2 NKExos efficiently targets Bcl-2, promotes enhanced apoptosis of breast cancer cells, and has better safety for normal cells [92]. Exosomes are also important in the treatment of colorectal cancer (CRC). In other studies, the anticancer drug 5-FU and miR-21 inhibitor oligonucleotide (miR-21) were also delivered into Her2-expressing cancer cells via engineered exosomes (Figure 3B). The results showed that it had a significant antitumor effect in tumor-bearing mice. Moreover, miR-21 expression in 5-FU-resistant HCT-116 5FR cell lines was significantly down-regulated. Fortunately, the combined delivery of miR-21i and 5-FU with engineered exosomes, when compared with miR-21i or 5-FU alone, achieved effective reversal of drug resistance and significantly enhanced 5-FU resistant colons cytotoxicity of cancer cells [93].

Exosomes have more prominent features than LNPs including nucleic acid delivery, protection of therapeutic substances from degradation, elimination by the host immune system, and etc. In addition, the characteristic of exosomes derived from their parental cells give them the potential for targeted delivery and enhance their ability to penetrate the tumor vascular barrier and bioaccumulate at tumor sites, greatly improving their therapeutic bottleneck. More importantly, exosomes have shown amazing feasibility as drug delivery vehicles in many preclinical studies and several clinical trials. From this point of view, the future of exosome-based delivery systems will break new ground in cancer treatment.

### 3.3. DNA Nanostructures

Tetrahedral framework nucleic acid (tFNA) has excellent mechanical, chemical, and biological properties, which make it a research hotspot in the field of DNA nanomaterials in recent years. Based on these advantages, tFNA has been designed and applied to the oncology therapeutic field [94,95]. DNA nanostructures, especially tetrahedral framework nucleic acids (tFNA), can also be used as drug delivery platforms to deliver RNA due to their cell-penetrating ability. Gao et al. constructed a pH-responsive DNA dynamic switch for RNA delivery [96] (Figure 4A,B). At pH = 7, the encapsulated siRNA in the scaffold is provided with undisturbed space because all strands forming the nanocapsules undergo regular complementary base pairing. At the acidity level of pH = 5, hemi-protonation and cytosine base pair (C: C+) were dominant, and C-rich DNA sequences tended to form the four-stranded structure of embedded motifs (I-motif). It was confirmed that these pH-responsive nanocapsules can promote the embedding of cargo siRNA across two key barriers: cell membrane and lysosomal membrane, and show considerable excellent gene silencing effects. In addition, the siRNA of Braf (siBraf) was modified with tFNAs (TFNAS-sibraF) at the sticky end to knock down the expression of target genes. Moreover, AS1411 (a DNA aptamer targeting nucleolins on the surface of A375 cells) was combined with TFNAS-siBraF (TFNAS-AS1411-SibraF) to further enhance cell uptake of nanostructures, which ultimately had a good inhibitory effect on malignant melanoma [27]. KerunLi reported a strategy to detect immunoadsorbent of rare tumor cells by signal amplification using CRISPR/Cas13a and tFNA [97] (Figure 4C). tFNA is used as a scaffold, which can effectively and stably anchor to the cell membrane and generate many stable sites for binding to target DNA. In addition, the sensitivity of TFNAS-Cas13A was significantly enhanced by dsDNA transcriptional amplification and activation of CRISPR/Cas13a cleavage activity by transcription products, which could reach the detection limit at the single-cell level. The study also tested clinical samples, further advancing the feasibility of TFNAS-Cas13A for clinical use.

DNA nanostructures have many outstanding features, including excellent biocompatibility and degradability, controllable self-assembly that endow them with structural precision and diversity, and the easy modification and introduction of functional groups, which have laid great potential for RNA delivery applications. But there are still huge problems for clinical application: such as (i) poor stability, (ii) immunogenicity, (iii) assembly efficiency and purification restricting its large-scale production.

### 3.4. Inorganic Nanoparticles

Compared with organic carriers, inorganic nanoparticles have other excellent physical and chemical properties, including: (i) stable size and shape, which can be precisely controlled in the synthesis process [98]; (ii) it can be combined with other polymer coatings to add desired capabilities to inorganic nanocarriers, such as biocompatibility and nucleic acid/drug loading [99]; (iii) when utilizing inorganic carriers as polymer-coated rigid carriers, loading and delivering nucleic acids without significantly changing the size or shape of the nanocarriers or even combining with other types of antitumor drugs to build multifunctional platforms [100]; (iv) the unique magnetic and optoelectronic properties, inorganic nanomaterials can also be used for additional treatments and imaging, such as sonodynamic therapy (SDT), photothermal therapy (PTT), photodynamic therapy (PDT) and X-ray-induced photodynamic therapy (XDT), as well as nuclear magnetic resonance imaging (MRI) and photoacoustic imaging (PAI) [101]. Thus, more and more researchers are using functional inorganic materials to deliver RNA.

Among the many carriers, calcium phosphate (CaP) nanoparticles have received special attention due to their superior physical and chemical properties and have brought broad prospects for siRNA loading. Kara et al. used arginine (Arg) modification to synthesize CaP nanoparticles with different chemical and morphological characteristics as vectors for survivin and Cyclin B1 specific siRNA (Figure 5A,B). The function of the CaP surface with Arg causes the surface to be positively charged. Functionalized nanoparticles show a higher loading capacity than unmodified particles [25]. In a subsequent study, Kara and his colleagues designed a gene carrier system based on superparamagnetic iron oxide nanoparticles (SPIONs) for siRNA delivery. SPIONs have the following advantages: (i) Unique superparamagnetism, which enables them to focus on targeted therapeutic sites where external magnetic fields are applied; (ii) they can be designed as a therapeutic diagnostic agent to achieve simultaneous treatment and diagnosis. However, the colloid of SPION is unstable, easily cleared in the blood circulation, and cytotoxic. Therefore, they designed a nanomaterial system composed of SPIONs and the biocompatible protein sericin (Ser), modified with the cationic polymer polylysine (PLL), that efficiently binds negatively charged siRNA [102] (Figure 5C). In the study of Cristofolini et al., caffeic acid-magnetic calcium phosphate (CAF-MCAP) nanoparticles with multifunctional magnetic nanostructure SPION were coated with caffeic acid and stabilized by layers of CaP and PEG-polyanion, for incorporation of siRNA to target breast cancer cells HER2 gene [103] (Figure 5D). In addition, various other inorganic materials, such as gold, silver, mesoporous silica (MS), iron oxide, graphene, titania, layered dihydroxide (LDH), etc., can be used as inorganic nanoparticle components to deliver RNA [21]. Therefore, the use of inorganic nanomaterials as carriers for RNA delivery has broad application prospects.

### 3.5. Lipid-Based Carriers

In 2018, the FDA and EMA approved the first lipid nanoparticle (LNP) based RNA interference (RNAi) drug of ONPATTRO TM (Patisiran), confirming that LNP is a valuable means of treating a variety of nucleic acid-related diseases. The use of LNP can overcome the limitations of naked siRNA in clinical applications, including their rapid degradation in biological fluids, limited biological distribution, and cellular absorption. However, factors of LNP lipid component diffusion and protein adsorption on the LNP surface caused the low potency. To alleviate these deficiencies, the strategy of using sphingomyelin (ESM) to replace the auxiliary lipid in cholesterol was proposed by Sato et al. The prepared LNP has a diameter of about 22 nm, and the gene silencing activity is greatly improved [104]. An iBL0713 lipid-based ionizable lipid nanoparticle (iLNP) was used for expressing desired proteins in vitro and in vivo using codon-optimized mRNA. ILP171/mRNA preparation was obtained by in vitro preparation of transcription of the mRNA encoding luciferase or erythropoietin (EPO) and preparation of the proposed iLNP [105]. Development of mRNA therapeutics was fueled by a useful and promising platform. A novel charge reversible lipid derivative diethylenediamine dioleoylglycerophosphate (DOP-DEDA) was synthesized (Figure 6A,B). DOP-DEDA LNP encapsulated siRNA targeting polo-like kinase 1 (siPLK1) highly inhibited PLK1 mRNA and protein expression. The results suggested that the LNP composed of reversible charge lipids should be a highly stable and efficient siRNA delivery vehicle, which can be widely used in the future [106]. Shobaki et al. constructed CL4H6 lipids composed of cholesterol (CHOL) and PEG and pristine pH-sensitive cationic lipids (Figure 6C). The vector effectively delivered siRNA to tumor-associated macrophages (TAM) and showed strong TAM gene silencing activity [107]. LNP loaded with mRNA consists of ionizable cationic lipids, phospholipids, cholesterol, and PEG lipids. Each component has its own integral role in efficient mRNA delivery and LNP stability. Under low pH conditions, ionizable lipids greatly facilitate cellular uptake and endosomal escape capacity by statically binding negatively charged mRNA core components during mixing. Xiong et al. designed a therapeutic diagnostic LNP platform combining biodegradable DLNP and PBD-lipids validated for efficient mRNA delivery in vitro and in vivo, and possessed non-invasive NIR imaging performance [108] (Figure 6D). Liposomes have the characteristics of high stability and remarkable uptake efficiency, but they cannot avoid the challenges of cytotoxicity and side effects.

### 3.6. Protein or Peptide as Carriers

Proteins or peptides are produced from various combinations of 20 naturally occurring amino acids with different 3-D conformations, charge, polarity, hydrophobicity, and hydrophilicity. These unique sequences can demonstrate a variety of properties, including siRNA binding, membrane penetration, endosomal destruction, targeting, etc. All these features are very helpful for RNA delivery. Moreover, the natural biological properties enable the direct screening of large proteomic libraries for a variety of interesting functional peptides [109].

Cell-penetrating peptides (CPPS) are amphiphilic oligomers with positively charged residues (arginine or lysine) composed of fewer than 30 amino acids that enable them to penetrate cells and drive a variety of organisms’ active substances (such as nanoparticles, RNA, proteins, or cytokines, etc.). In most cases, the natural helical folding properties of some CPPS when interacting with biological membranes assist improve the intracellular delivery of peptides. Simon’s team have carefully designed and synthesized alpha-helical peptide sequences consisting of short helical peptides with only eight amino acids and two cationic charges, and can efficiently encapsulate siRNA and induce gene silencing in the nanomolar range [110] (Figure 7A,B). Xu et al. designed a novel siRNA vector consisting of protamine and aptamer, known as aptamer-protamin-siRNA nanoparticles (APR) (Figure 7C). In this nanoparticle, protamine is a cationic peptide that binds strongly to nucleic acids. Protamine has been used as a stabilization carrier for RNA delivery due to its cationic nature [111]. In general, protein drugs have the advantages of higher specificity, greater activity, and lower toxicity than traditional small molecule drugs [28]. Cell-surface-specific receptors (such as ion channels or G protein-coupled receptors to which signaling molecules bind) are influenced by the protein’s specific affinity and greater activity, capable of inducing intercellular effects [112]. A strategy for encapsulating siRNA into EV using a polycation membrane penetrating peptide TAT was reported by Diao et al. The three tat-peptides and DRBD were expressed as 3TD fusion proteins. Sequence-independent binding of DRBD promotes multi-gene targeting of mixed siRNA [113] (Figure 7D). Protein or polypeptide delivery RNA strategies are highly selective and active, but the production cost and complex process limit its expansion.

## 4. RNA Delivery in Clinical Practice

As a more precise and effective emerging treatment strategy, gene therapy has gradually emerged in the field of clinical application in the past two decades. Several RNA-based products have been successfully used in the clinic, and many more are in various stages of development [114]. The applications of RNA include encoding disease-related proteins, silencing protein expression of specific genes, regulating protein function, and mediating gene transcriptional activation. In the development of nucleic acid drugs, although a variety of drugs have been approved for marketing, RNA drugs for cancer treatment are still in the early stage of clinical trials, as illustrated below (Table 1).

RNAi-based nucleic acid drugs have been used in many clinical trials to inhibit tumorigenic genes and inhibit cancer development. Most clinical drugs use liposome nanoparticles as siRNA delivery vectors. For example, nucleic acid drugs such as Atu027, TKM-080301, and EPHARNA are all delivered by liposome nanoparticles. Local Drug EluteR is a miniature biodegradable polymer matrix that is well tolerated and safe in combination with chemotherapy encapsulated in the LODER extended delivery system (targeting KRASG12D), showing significant efficacy in the treatment of pancreatic cancer [115]. There is also an ongoing clinical study using mesenchymal stromal cell-derived exosomes as vectors for the treatment of metastatic pancreatic cancer with KRASG12D siRNA (NCT03608631). In addition, inanimate bacterial microcells (EDVs) act as leak-proof micro-reservoir carriers, allowing efficient packaging of a range of different drugs, proteins, or nucleic acids. In this clinical study, Mir-16-based microRNA mimics coated with EDV demonstrated the properties of egFR-expressing cancer cells targeted by anti-EGFR-specific antibodies (NCT02369198).

In addition to RNAi-based therapies, mRNA anticancer vaccines have made significant clinical advances. mRNA drugs currently under clinical investigation for cancer treatment have vaccine-like properties, promoting the production of tumor-specific and tumor-associated antigens on the surface of cancer cells, further activating the host immune system. A particular advantage of mRNA cancer vaccines is that they can be customized to the patient, thereby improving the efficacy of the vaccine. The mRNA-4157 vaccine, an mRNA vaccine developed by Moderna, uses a novel gene-based technique to identify tumor-specific changes to DNA by comparing the normal cell DNA sequence of a patient with the normal DNA sequence of the tumor. By combining with pembrolizumab, the researchers hypothesized that the vaccine could trigger the immune system to respond more sensitively to PD-1 inhibitors and reduce the risk of cancer recurrence.

Although the first generation of RNA drugs have been approved for clinical use and more are in the development, the long-term effects of RNA therapy are still unclear, and some clinical trials have even been stopped because of unclear mechanisms. The safety risks of RNA therapy are the important issues [116]. Although RNA-mediated cancer treatment strategies are still in their infancy, their high specificity and transient nature give RNA therapies unique properties that are still worth exploring in cancer treatment.

## 5. Conclusions and Future Prospects

In this review, we summarized RNA therapeutic-based nucleic acid delivery strategies for cancer therapy. After decades of development of RNA therapeutics, the targeting and delivery efficiency have improved significantly. RNA therapies alter the expression of target genes by regulating internal cellular mechanisms, which offers a wide range of potential applications and provides a high degree of flexibility to manipulate previously “unusable” targets. Therefore, siRNA can modulate the expression of various genes which has raised prospect for the new therapeutic strategies. Nowadays, RNA-based cancer treatment strategies are receiving increasing attention, mainly due to the flexibility of targeting previously unmanageable disease genes and modulating a wide range of targets. As summarized in the manuscript, we highlight effective strategies for RNA delivery by different types of carriers (cationic polymer, inorganic nanoparticles, lipids, etc.), and describe the advantages and disadvantages of each. Meanwhile, the clinical translation progress, challenges, and application prospects of RNA in cancer therapy were reviewed. Although various RNA drugs have been developed, only a few have completed phase 1 clinical trials, clinical application will take more time due to the challenges of nucleic acid stability and unclear delivery strategies and mechanisms.

An ideal nanocarrier for RNA delivery should meet the conditions of high safety, biocompatibility, biodegradability, and a high gene transfection rate. Advances in nanotechnology will help address these issues, and nanomaterials hold promise as a potential delivery vector to address the challenges and drawbacks of RNA therapeutics. With the application and development of lipid-based nanoparticles in the clinical scenario of COVID-19 mRNA vaccines, nanotechnology has shown great effectiveness and availability in RNA delivery strategies. According to the previous summary, an effective RNA nanodelivery vector formulation should have these characteristics: (i) low toxicity and serum-stable RNA loading methods, (ii) accurate targeting of target genes, and (iii) a high degree of uptake by target cells and endosomal escape. Through simple structural modification and packaging of the basic nanomaterial carrier, such as the use of specific cell membrane vesicles to make it targeted and biosafe, the carrier can achieve multifunctional requirements containing the above characteristics. The optimization of nanomaterials will provide a safe and effective method of RNA delivery, which will make RNA have a broader application prospect in cancer treatment.

Although there are still some unsolved problems, it is believed that with the deepening of research, the continuous improvement of material understanding, and the increasing exploitation of new materials, safe and efficient RNA delivery solutions will be applied to the clinic in the near future. It is also hoped that RNA therapy will be more widely adopted as a new class of therapies in combination with other therapies or drugs. At present, the mechanism of RNA therapy is also constantly revealed and explored, which is expected to shine in cancer treatment.

## Figures and Tables

**Figure 1 bioengineering-09-00576-f001:**
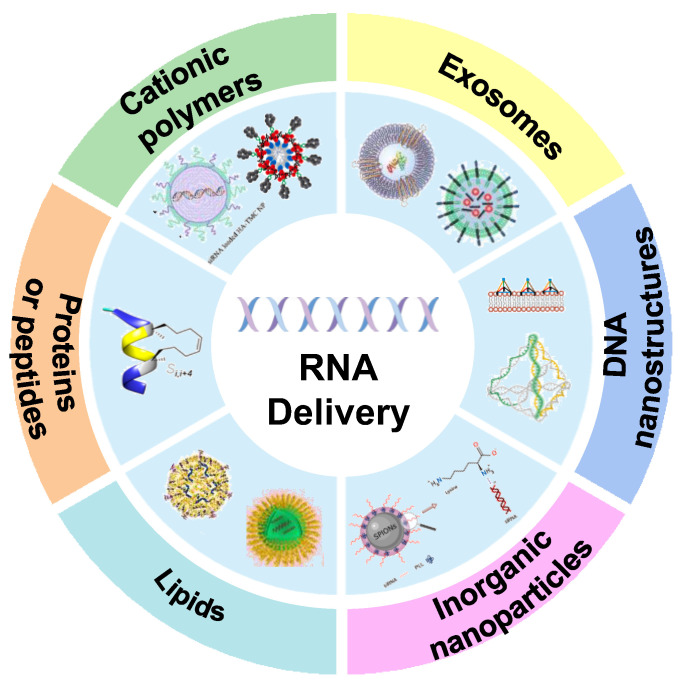
Different RNA delivery carriers in cancer therapy. These delivery vehicles include cationic polymers, exosomes, DNA nanostructures, inorganic nanoparticles, lipids, proteins or peptides, etc.

**Figure 2 bioengineering-09-00576-f002:**
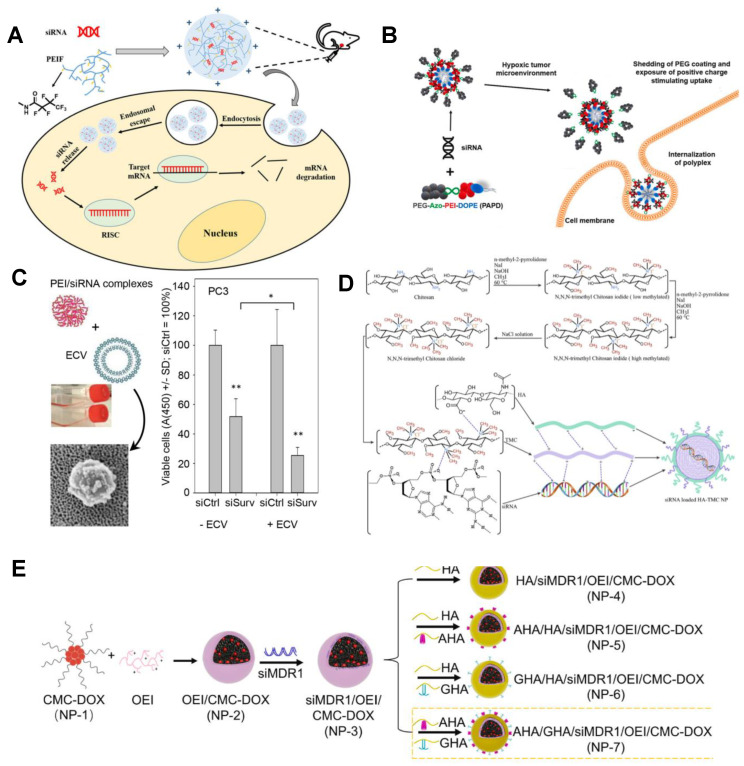
(**A**) The formation process of PEIF/siRNA complex and its application in tumor cells. Fluorinated PEIF was synthesized by forming an amide bond between PEI and heptafluorobutyric anhydride. The complex system formed by the combination of PEIF and nucleic acid is stable and easy to be taken up by cells, with high endosomal escape efficiency and gene silencing ability [66]. Copyright 2020, Springer Nature. (**B**) Schematic illustration shows the PAPD-siRNA and its mode of action in tumor cells. When PAPD nanoparticles are extravasated into tumor tissue, the hypoxic microenvironment causes bio-reducing cleavage of azo adaptor in the activated PAPD complex, resulting in shedding of PEG layer and exposure of the positive charged PEI previously shielded by PEG. This leads to increased uptake of nanoparticles by tumor cells [67]. Copyright 2020, Elsevier. (**C**) Production and therapeutic effect of PEI/siRNA nanoparticles modified by ECVs. ECV-modified PEI/siRNA complex targeting survivin increased inhibition of PC3 prostate cancer cells Tumor growth curves showed that tumor growth was significantly inhibited by approximately 45% in the treated group [68]. Copyright 2020, Elsevier. (**D**) TMC synthesis steps and schematic diagram of siRNA loaded HA-TMC NP [26]. Copyright 2020, Elsevier. (**E**) The fabrication process of siMDR1/DOX co-delivery nanosystems. The nanoplatform consists of three components: the redox-responsive and negatively charged CM-DOX “core”, the pH-responsive dissociated and positively charged “shell” oligoethylene imide /siMDR1, and the surface-modified AS1411 aptam-conjugated hyaluronic acid (AHA) and GALA peptide-conjugated hyaluronic acid (GHA) [69]. Copyright 2022, Elsevier. * *p* < 0.05, ** *p* < 0.01.

**Figure 3 bioengineering-09-00576-f003:**
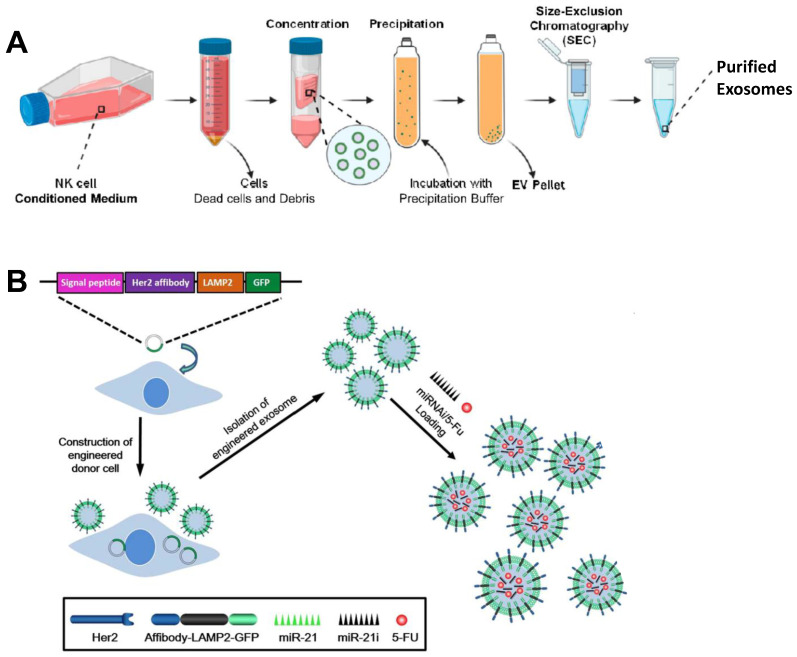
(**A**) The diagram describes the NKExos production process diagram. After 72 h of culture, NK92MI cells were removed by differential centrifugation. Finally, NKExos was concentrated and purified [92]. Copyright: © 2021 by the authors. (**B**) Preparation flow chart of engineered exosome nanocapsules based on 5-FU and Mir-21i. Her2 fuses with LAMP2 to form HCT-116 5FR cells. Her2-lamp2 fusion protein promotes targeted cellular uptake through EGFR receptor-mediated endocytosis in HCT-116 cells. 5-FU and Exo were mixed by electroporation, and 5-FU and Mir-21i were packaged into engineered exosomes and incubated together to form a co-delivery system (THLG-EXO/5-FU/Mir-21I) [93]. Copyright 2020, Springer Nature.

**Figure 4 bioengineering-09-00576-f004:**
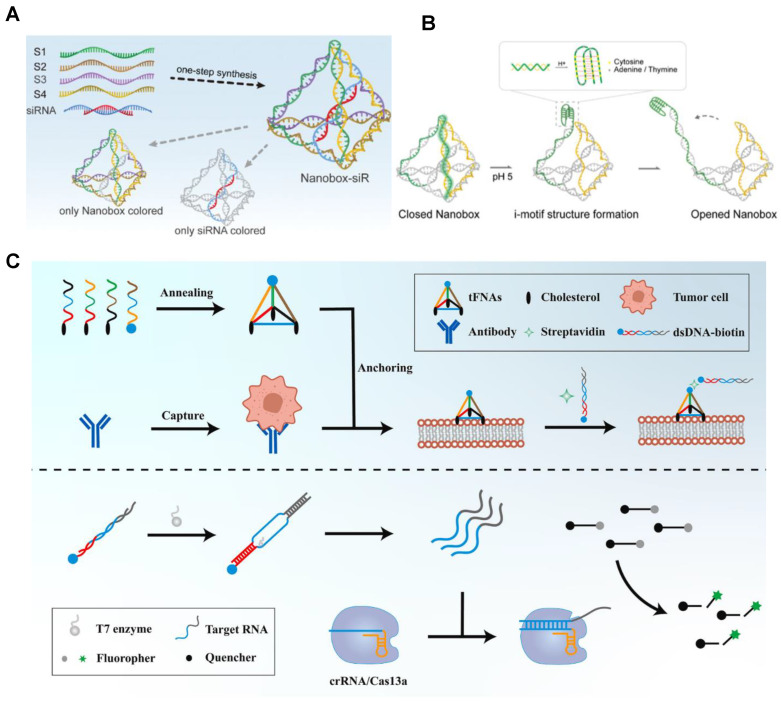
Schematic diagram of the nanobox-siR synthesis (**A**) and structural transition of its pH-responsive switch (**B**). The tetrahedral nanobox exoskeleton and siRNA encapsulation and protection were formed simultaneously by one-pot annealing (95 °C for 10 min and 4 °C for 20 min). At pH = 5, the nanobox nucleic acid sequence is transformed into a four-stranded I-motif structure, which subsequently leads to structural disintegration of the spatial tetrahedron and release of the target siRNA [96]. Copyright 2022, Wiley-VCH. (**C**) Schematic diagram of tFNAS-Cas13a synthesis process and its action on cancer cells. These cells were surface engineered using amphipathic tFNA with three cholesterol-modified vertices and one biotin-modified vertex. T7 polymerase recognizes dsDNA in the T7 promoter region and simultaneously generates a large number of copies of single-stranded RNA. Cas13a recognizes target RNA to activate CRISPR/Cas13a trans cleavage capability, and ssRNA with a fluorophore and quench groups is cleaved by this RNA cleavage capability [97]. Copyright 2022, Elsevier.

**Figure 5 bioengineering-09-00576-f005:**
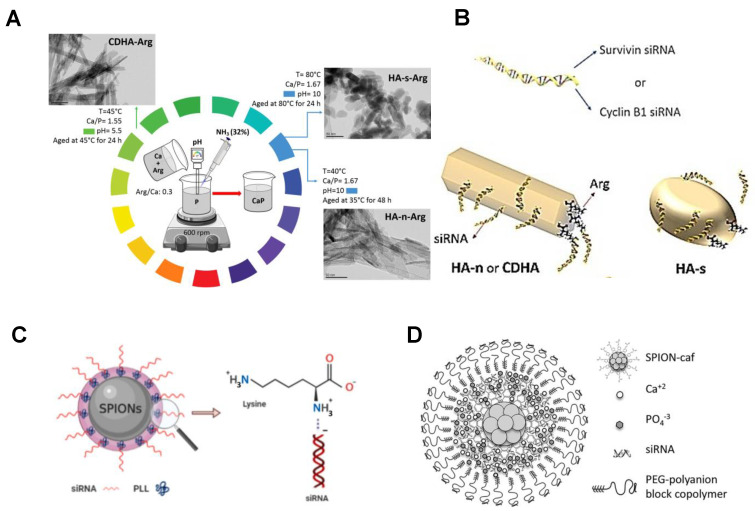
(**A**) Schematic diagram of CaP nanoparticles with different chemical and morphological characteristics and a simplified diagram for the delivery of survivin and cyclin B1-specific siRNA. (**B**) Through this procedure, acicular hydroxyapatite (HA-N), spherical hydroxyapatite (HA-S) and calcium-deficient hydroxyapatite (CDHA) with average particle sizes of 15, 38, and 20 nm were obtained. The aspect ratio of HA-N and CDHA is about 6. CaP-Arg-siRNAs effectively down-regulated the expression of survivin and cyclin B1 genes to significantly induce cell apoptosis [25]. Copyright 2020, Elsevier. (**C**) Schematic diagram of gene carrier system of superparamagnetic iron oxide nanoparticles (SPIONs). The consists of SPION coated with the biocompatible protein sericin (Ser) and modified with the cationic polymer polylysine (PLL) to bind negatively charged siRNA [102]. Copyright 2021, Elsevier. (**D**) Structure diagram of caffeic acid-magnetic calcium phosphate (CAFA-MCAP) nanoparticles. The nucleus consists of superparamagnetic iron oxide nanoparticles (SPION) coated with caffeic acid and stabilized by calcium phosphate, siRNA, and PEG-polyanionic block copolymer layers [103]. Copyright 2020, Elsevier.

**Figure 6 bioengineering-09-00576-f006:**
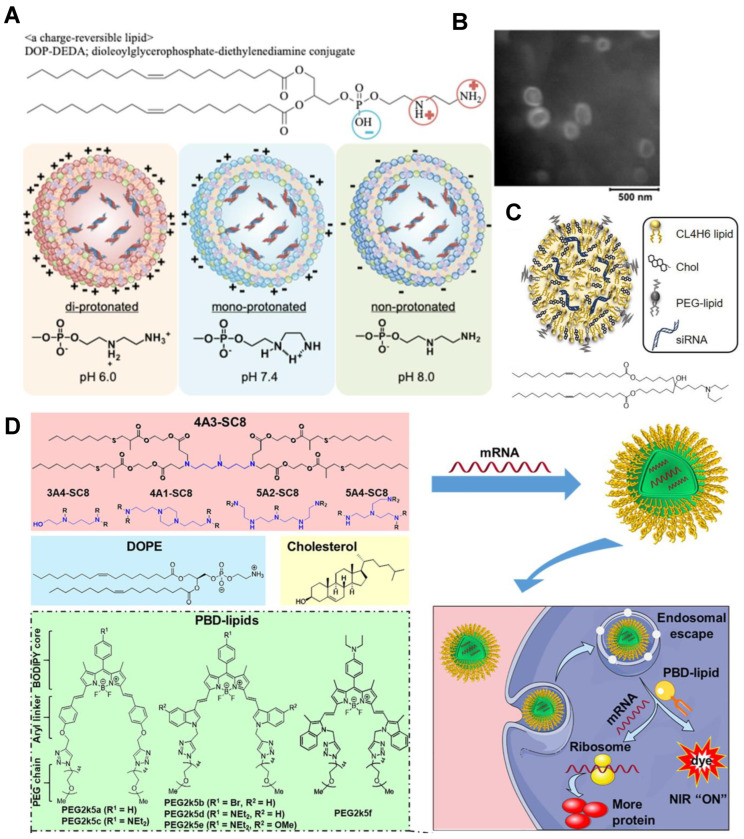
(**A**) Schematic diagram of the DOP-DEDA structure and pH response of DOP-DEDA LNPs, and TEM image (**B**) of DOP-DEDA LNP. DOP-DEDA has a negatively charged phosphate and two amino groups linked by an ethylene bridge. Produces an almost neutral charge at pH 7.4. The total positive charge at pH = 6.0 and the total negative charge at pH = 8.0 [106]. Copyright 2020, Elsevier. (**C**) LNPs nanostructure and CL4H6 lipid chemical structure. LNP is composed of CL4H6 lipids, CHOL and PEG-lipids [107]. Copyright 2020, Elsevier. (**D**) Schematic representation of dendrimer/DOPE/cholesterol/PBD-lipid/mRNA nanoparticles and mRNA delivery process. The combination of PDB lipids and DLNP enhances mRNA production in cancer cells, illuminating tumors by PH-responsive NIR imaging [108]. Copyright 2020, Elsevier.

**Figure 7 bioengineering-09-00576-f007:**
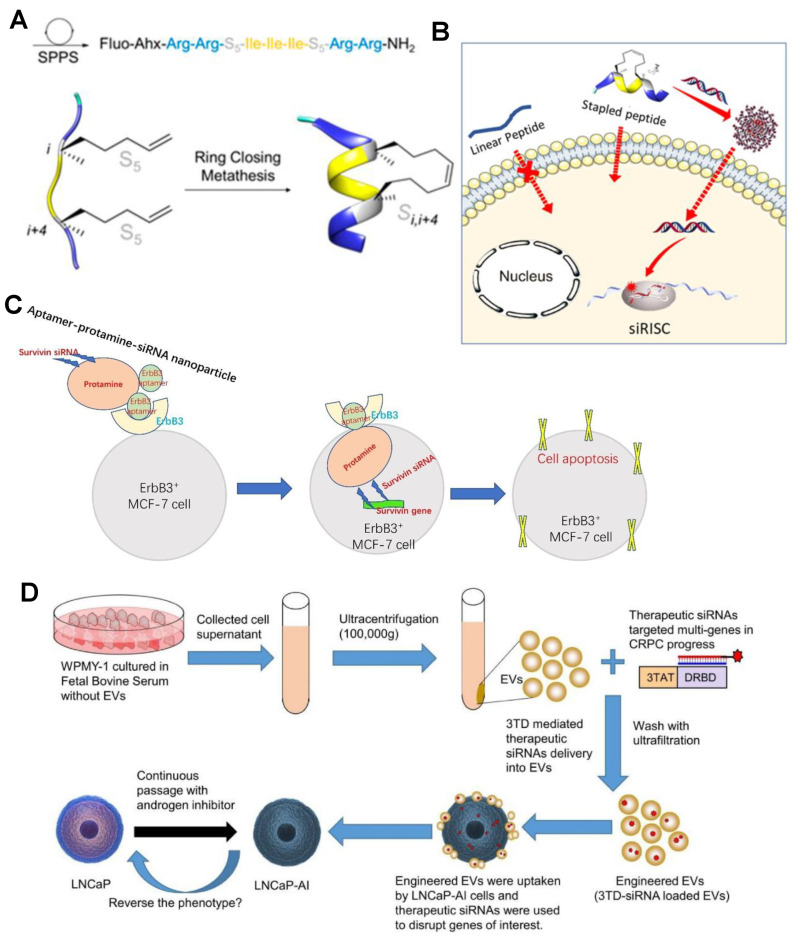
(**A**) Peptide synthesis and gene silencing process (**B**). Fluorescent pin peptides were synthesized by closed-loop metathesis (JMV6337) to encapsulate siRNA and induce gene silencing using short helical pin peptides [110]. Copyright 2020, MDPI. (**C**) Schematic diagram of APR nanoparticle composition and silencing targeted gene expression. APR nanoparticles are composed of ErbB3 aptamer, protamine and siRNA. After entering cells by ErbB3 aptamer recognition, siRNA silenced the survivin gene and induced apoptosis of ErBB3-positive cells [111]. Copyright 2020, Elsevier. (**D**) Schematic diagram of EV delivery strategy of siRNA and predicted therapeutic effect on CRPC cells [113]. Copyright 2022, Informa UK Limited.

**Table 1 bioengineering-09-00576-t001:** RNA drugs in clinical practice.

RNA Drug Name	Target	Type of RNA	Delivery System	Cancer Types	Phase	ClinicalTrials.gov Identifier	Reference
**Atu027**	PKN3	siRNA	Cationic LNPs	Advanced or metastatic pancreatic cancer	I/II	NCT00938574 /NCT01808638	[117,118]
**siG12D LODER**	KRAS	siRNA	Polymeric NPs (LODER)	Pancreatic ductal adenocarcinoma, pancreatic cancer	I/II	NCT01188785/ NCT01676259	[119]
**Mesenchymal stromal cells-derived exosomes with KRAS G12D siRNA**	KRAS G12D	siRNA	Exosomes	Pancreatic cancer	I	NCT03608631	[120]
**TKM-080301**	PLK1	siRNA	LNPs	Adrenal cortical carcinoma, neuroendocrine tumor, hepatocellular carcinoma	I/II	NCT01262235 /NCT01437007 /NCT02191878	[121,122]
**EPHARNA**	EphA2	siRNA	LNPs	Advanced malignant solid neoplasm	I	NCT01591356	[123]
**NU-0129**	BCL2L12	siRNA	Au NPs	Gliosarcoma, recurrent glioblastoma	I	NCT03020017	[124]
**ALN-VSP02**	VEGF and KSP	siRNA	SNALPs	Solid tumors	I	NCT01158079	[125]
**TargomiRs**	Mir 16	miRNA	nonliving bacterial minicells (nanoparticles)(EDVs)	Malignant pleural mesothelioma, Non-small cell lung cancer	I	NCT02369198	[126]
**mRNA-4157 and Pembrolizumab**	Individually designed mRNA coding for tumor neoantigens	mRNA	LNP	High-risk melanoma	II	NCT03897881 /NCT03313778	[127]
**mRNA-2416 and Durvalumab**	OX40	mRNA	Liposomes (SM-102, DSPC, PEG2000-DMG, cholesterol)	cancer	I/II	NCT03323398	[128]
**Lipo-MERIT**	Antigen-specific differentiated clusters of four selected malignant melanoma-associated antigens	mRNA	LNP	Melanoma	I	NCT02410733	[129]
**MTL-CEBPA**	C/EBPα	saRNA	LNP	Liver cancer	I	NCT02716012	[130]

## Data Availability

Not applicable.

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
