# Peer review of "Broadening the Horizons of RNA Delivery Strategies in Cancer Therapy"

_bioengineering, 2022, doi:10.3390/bioengineering9100576_

Round 1

Reviewer 1 Report

This review summarized current mechanisms of RNA-based therapy, excellently. There is a few points should be revised.

Major points.

#1. Figures contain many citations from another group’s publication. Please prepare your original art works to minimize the possibility of copyright violation.

#2. In Figure 3(A), authors cited TEM image from 70, which is published by another group in MDPI journal. It may not necessary.

#3. In Figure 3(B), authors cited TEM images from reference 71 from Springer Nature’s journal. This may not necessary.

Author Response

Reviewer #1: This review summarized current mechanisms of RNA-based therapy, excellently. There are a few points should be revised.

Response: We appreciate the reviewer for the positive comments and recognition of our work, and put forward detailed suggestions for further improving the quality of the manuscript. We carefully revised and addressed each question point-by-point.

  1. Figures contain many citations from another group’s publication. Please prepare your original art works to minimize the possibility of copyright violation.

Response: Thank you for reminding us that we have applied for copyright for all the quoted pictures.

  1. In Figure 3(A), authors cited TEM image from 70, which is published by another group in MDPI journal. It may not necessary.

Response: Thanks for your reminder. As suggested, we have modified Figure 3(A) by removing TEM image from reference 70.

  1. In Figure 3(B), authors cited TEM images from reference 71 from Springer Nature’s journal. This may not necessary.

Response: We thank this reviewer for the valuable suggestion and we have modified Figure 3(B) by removing TEM image from reference 71.

Reviewer 2 Report

The manuscript entitled “Broadening the horizons of RNA delivery strategies in cancer therapy” by Wu et al. provides a comprehensive overview of various carriers of RNA and strategies for RNA delivery. Besides classification of RNA carriers, the authors consider different strategies of using RNAi and mRNA as well as RNA delivery in clinical practice by giving many examples from the recent scientific literature. Challenges, drawbacks, limitations as well as future development and prospects of RNA delivery are critically discussed. I have some reservations about using singular/plural, particularly for the titles of section 3: Classification of RNA drug carrier. Is the carrier only one or they are many carriers and can be classified in different groups? The same applies for 3.1 Cationic polymer (only one polymer?), 3.2 Exosome, 3.5 Lipid-based carrier, and 3.6 Protein or peptide carrier. Furthermore, I do not understand why chitosan, which is a cationic polymer, is given and considered separately (see Figure 1, the text above Figure 1, section 3.7) rather than in the section for the cationic polymers (section 3.1). In the latter section (section 3.1) polyethyleneimine (PEI) is wrongly named as polyvinyl imine, whereas its monomer unit is called “ethylamine” (page 4, section 3.1). The authors have to consider combining section 3.1 and section 3.7 into one section “Cationic polymers”.

There are many incomplete and meaningless sentences. Below are some examples:

The outstanding features of exosomes over LNPs, such as low immunogenicity, low toxicity, and ability of cross biological barriers. (page 7, the last paragraph)

The voluntary release of cargo from a protective carrier to overcome transportation obstacles. (page 8, the beginning of section 3.3)

In addition, various other inorganic materials, such as gold, silver, mesoporous silica (MS), iron oxide, graphene, titania, layered dihydroxide (LDH), etc., which can be used as inorganic nanoparticle components to deliver RNA [13]. (page 10, the bottom lines)

CL4H6-LNP for efficient targeting and delivery of siRNA to TAM, which mainly consists of three components, CL4H6, CHOL, and DSG-PEG 2000 (60/40/1 mol % of total lipids) [85] . (page 12)

There are many other sentences like those that need to be inspected and revised.

Some minor points are:

Page 4, first paragraph of section 3.1, about the buffering capacity and proton sponge effect it is stated that “RNA is protonated in endosomal vesicles under acidic conditions” but, actually, it is the polymer, not nucleic acids, that is protonated under acidic conditions. Some lines below, it is better to use “linear” rather than “straight” chain in the discussion of the chain architecture of polyethyleneimine. In the figure caption of Figure 2B probably the positive charges are shielded by PEG, not by PEI as it is written.

Based on the above, I recommend major revision and extensive language editing of the manuscript before accepting for publication.

Author Response

The manuscript entitled “Broadening the horizons of RNA delivery strategies in cancer therapy” by Wu et al. provides a comprehensive overview of various carriers of RNA and strategies for RNA delivery. Besides classification of RNA carriers, the authors consider different strategies of using RNAi and mRNA as well as RNA delivery in clinical practice by giving many examples from the recent scientific literature. Challenges, drawbacks, limitations as well as future development and prospects of RNA delivery are critically discussed. I have some reservations about using singular/plural, particularly for the titles of section 3: Classification of RNA drug carrier. Is the carrier only one or they are many carriers and can be classified in different groups? The same applies for 3.1 Cationic polymer (only one polymer?), 3.2 Exosome, 3.5 Lipid-based carrier, and 3.6 Protein or peptide carrier. Furthermore, I do not understand why chitosan, which is a cationic polymer, is given and considered separately (see Figure 1, the text above Figure 1, section 3.7) rather than in the section for the cationic polymers (section 3.1). In the latter section (section 3.1) polyethyleneimine (PEI) is wrongly named as polyvinyl imine, whereas its monomer unit is called “ethylamine” (page 4, section 3.1). The authors have to consider combining section 3.1 and section 3.7 into one section “Cationic polymers”.

Response: We appreciate the reviewer’s rigorous attitude to our manuscripts and apologize for this syntax problem and have corrected the singular and plural. Thank you for pointing this out. We agree that it makes more sense to use the plural number because the carrier mentioned is a category, not just a type of carrier. For another question, we would also like to thank you for your suggestion. Since chitosan is a natural cationic polymer and cationic polymers in 3.1 are all synthetic, we made a separate classification, but we ignored that they all belong to the type of cationic polymer. We are very sorry for the separate classification of chitosan due to our negligence. We have merged 3.7 into 3.1 according to your suggestion and modified Figure 1 accordingly. Thank you again for the important suggestions. Here, the classification of RNA drug carrier was divided into six types for discussion.

In the latter section (section 3.1) polyethyleneimine (PEI) is wrongly named as polyvinyl imine, whereas its monomer unit is called “ethylamine” (page 4, section 3.1).

Response: We thank this reviewer for the valuable suggestion and we have corrected these errors.

There are many incomplete and meaningless sentences. Below are some examples:

Response: Thank you for your advice. We are sorry that our language is not clear. According to your suggestion, we made changes one by one to make the text more fluent, and carefully examined the other sentences and revised the whole text.

  1. The outstanding features of exosomes over LNPs, such as low immunogenicity, low toxicity, and ability of cross biological barriers. (page 7, the last paragraph)

Response: Exosomes have more prominent features than LNPs including nucleic acid delivery, protection of therapeutic substances from degradation, elimination by the host immune system, and etc.

  1. The voluntary release of cargo from a protective carrier to overcome transportation obstacles. (page 8, the beginning of section 3.3)

Response: We have removed this sentence.

  1. In addition, various other inorganic materials, such as gold, silver, mesoporous silica (MS), iron oxide, graphene, titania, layered dihydroxide (LDH), etc., which can be used as inorganic nanoparticle components to deliver RNA [13]. (page 10, the bottom lines)

Response: In addition, various other inorganic materials such as gold, silver, mesoporous silica (MS), iron oxide, graphene, titania, and layered dihydroxide (LDH), can also be used as components of inorganic nanoparticles to deliver RNA [21].

  1. CL4H6-LNP for efficient targeting and delivery of siRNA to TAM, which mainly consists of three components, CL4H6, CHOL, and DSG-PEG 2000 (60/40/1 mol % of total lipids) [85]. (page 12)

Response: The vector effectively delivered siRNA to tumor-associated macrophages (TAM) and showed strong TAM gene silencing activity.

  1. There are many other sentences like those that need to be inspected and revised.

Response: We appreciate the reviewer’s rigorous attitude to our manuscripts. We have made a thorough examination of the full text to ensure that such errors are eliminated.

  1. Some minor points are:

(1)Page 4, first paragraph of section 3.1, about the buffering capacity and proton sponge effect it is stated that “RNA is protonated in endosomal vesicles under acidic conditions” but, actually, it is the polymer, not nucleic acids, that is protonated under acidic conditions.

Response: Thank you very much for asking such a rigorous question. We are sorry for your misunderstanding because we did not express ourselves clearly. We have made changes to the corresponding parts of the manuscript to make the expression clearer and more accurate. At the same time, this part of the content is explained as follows: this part is to show that the proton buffering capacity of cationic polymer contributes to the endosomal escape of RNA. RNA is negatively charged because it contains a large number of phosphate groups, while acidic environments contain a large number of hydrogen ions, so negatively charged RNA will combine with hydrogen ions and undergo protonation, eventually leading to a series of subsequent reactions and successful endosomal escape.

(2)Some lines below, it is better to use “linear” rather than “straight” chain in the discussion of the chain architecture of polyethyleneimine.

Response: Thank you very much for finding this error. We apologize for this grammar problem and have corrected it according to your suggestion. We have replaced “straight” with “linear”.

(3)In the figure caption of Figure 2B probably the positive charges are shielded by PEG, not by PEI as it is written.

Response: Thank you for finding this error. We apologize for this problem and have modified it accordingly.

(4)Based on the above, I recommend major revision and extensive language editing of the manuscript before accepting for publication.

Response: Thank you for such valuable advice. We apologize for the poor language of our manuscript.  We worked on the manuscript for a long time and the repeated addition and removal of sentences and sections obviously led to poor readability. We have now worked on both language and have carefully polished and revised the manuscript in accordance with your suggestion. We really hope that the flow and language level have been substantially improved.

Reviewer 3 Report

In the paper entitled “ Broadening the horizons of RNA delivery strategies in cancer therapy”, the author discusses delivery obstacles in RNA-mediated cancer therapy and gives a comprehensive overview of various carriers and delivery strategies for RNA delivery. Meanwhile, the status of clinical applications and practice of RNA medicines are classified and integrated. This paper has clear logic and comprehensive content, enabling people in related fields or non-related fields to quickly understand this field by reading this article. 

There are some problems, which must be considered in subsequent versions:

Comment 1: Relevant research background needs to be supplemented in introduction. 

The introduction of this article lacks the background of RNA as a genetic medicine, including its discovery and development process.

Comment 2: Some sentences contain mistakes 

1.      In page 3, the abbreviation was used directly for the first use of “mRNA”

2.      In page 3, “micro RNAs (miRNAs)” and “these non-coding small RNA molecules (miRNAs)”, why there are two different definitions for (miRNA)?

3.      In page 4, “Polyvinyl imine, as a wwell-known polymer…”, there is an extra “w”

4.      In page 19, the last paragraph, “… and nanomaterials or nanomaterials hold…”, there are two “nanomaterials”

Comment 3: Add a review of nanomaterials as RNA delivery carriers.

As you mentioned, “nanomaterials hold promise as a potential delivery vector to address the challenges and drawbacks of RNA therapeutics”. However, the article does not discuss nanomaterials as carriers, such as nanotubes or nanosheets. This part should be promising and attractive. Hope the author can add this part to make the article more complete.

Comment 4: “Future prospects” should be more in-depth

After reviewing the articles in this field and combining with your own thinking, the author should put forward some forward-looking suggestions for this field, which can guide the work in this field. This is an important distinction between your article and similar review articles. As a reader, I hope to get some inspiration.

Author Response

In the paper entitled “Broadening the horizons of RNA delivery strategies in cancer therapy”, the author discusses delivery obstacles in RNA-mediated cancer therapy and gives a comprehensive overview of various carriers and delivery strategies for RNA delivery. Meanwhile, the status of clinical applications and practice of RNA medicines are classified and integrated. This paper has clear logic and comprehensive content, enabling people in related fields or non-related fields to quickly understand this field by reading this article.

  1. Relevant research background needs to be supplemented in introduction.

The introduction of this article lacks the background of RNA as a genetic medicine, including its discovery and development process.

Response: Thank you for your valuable suggestions. We have revised it according to your suggestions, and added the development and history of RNA therapy in the introduction, hoping to make the content of the article more complete and more fluent. We added the following: In 1958, Crick's central law was the first to mention that RNA plays a crucial role in the transmission of genetic information [12]. In subsequent studies, RNA was found to be capable of complementary base pairing to form a double-stranded structure similar to DNA, which laid a foundation for the discovery of mRNA-interfering complementary RNA (micRNA) and the emergence of RNA interference drugs [13]. Complementary pairing of RNA bases was first used to treat diseases in 1978 [14]. The RNA interference strategy was first described in 1998 in the study of Andrew et al [15]. Two decades later, the first siRNA drug to treat patients with hereditary thyroxine transporter-associated amyloidosis was approved. In addition to the therapeutic use of RNA interference, messenger RNA (mRNA) is also used in the prevention and treatment of diseases. An mRNA vaccine for cancer treatment was first designed in 1995 [16]. In 2008, the results of the first clinical trials of mRNA vaccines were reported [17]. Most recently, the first mRNA vaccine against SARS-CoV-2 was approved in 2020 [18].

  1. Some sentences contain mistakes
  2. In page 3, the abbreviation was used directly for the first use of “mRNA”

Response: Subsequently, siRNA binds to the polymerase complex (RISC) and performs the effector function of RNA interference through the corresponding messenger RNA (mRNA) site.

  1. In page 3, “micro RNAs (miRNAs)” and “these non-coding small RNA molecules (miRNAs)”, why there are two different definitions for (miRNA)?

Response: Thank you for pointing out the mistake and we are sorry that it was caused by our carelessness in writing. Non-coding RNA (ncRNA), ncRNA mainly includes microRNA (miRNA), small interfering RNA (small interfering RNA), siRNA), PiWI-Interacting RNA (piRNA), trNA-derived Small RNA (tsRNA), Small nuclear RNA (tsRNA), SnRNA, small nucleolar RNA (snoRNA), long non-coding RNA (lncRNA) and circular RNA (circRNA). Here is an abbreviation error, the “miRNA” in parentheses has been modified to “ncRNA”.

3.In page 4, “Polyvinyl imine, as a wwell-known polymer…”, there is an extra “w”

Response: Thank you for pointing out the mistake and we are sorry that it was caused by our momentary negligence. We have made changes to remove the superfluous characters.

4.In page 19, the last paragraph, “… and nanomaterials or nanomaterials hold…”, there are two “nanomaterials”

Response: We sincerely thank you for pointing out the error. We have revised it to remove the superfluous words.

  1. Add a review of nanomaterials as RNA delivery carriers.

As you mentioned, “nanomaterials hold promise as a potential delivery vector to address the challenges and drawbacks of RNA therapeutics”. However, the article does not discuss nanomaterials as carriers, such as nanotubes or nanosheets. This part should be promising and attractive. Hope the author can add this part to make the article more complete.

Response: Thank you for your suggestion and we have thought it over. But we think that the types of nanostructures such as the nanosheets and nanotubes that you mentioned, these are a structural form of the materials, not the type of carriers themself. All the carrier types we mentioned have the possibility to be prepared into nanomaterials, such as nanoparticles, nanorods or nanosheets. These may not be the focus of this paper, and we may discuss them separately in a future paper.

  1. “Future prospects” should be more in-depth

After reviewing the articles in this field and combining with your own thinking, the author should put forward some forward-looking suggestions for this field, which can guide the work in this field. This is an important distinction between your article and similar review articles. As a reader, I hope to get some inspiration.

Response: Thank you for your constructive suggestions, and we are sorry that our previous outlook was not in-depth enough. According to your suggestions, we added the importance of improving the delivery strategy of nanomaterials as a carrier and suggestions for transformation in the future outlook of the last part, hoping to enrich the content and readability of the manuscript. We added the following: An ideal nanocarrier for RNA delivery should meet the conditions of high safety, good biocompatibility, biodegradability, and a high gene transfection rate. Advances in nanotechnology will help address these issues, and nanomaterials hold promise as a potential delivery vector to address the challenges and drawbacks of RNA therapeutics. With the application and development of lipid-based nanoparticles in the clinical scenario of COVID-19 mRNA vaccines, nanotechnology has shown great effectiveness and availability in RNA delivery strategies. According to the previous summary, an effective RNA nanodelivery vector formulation should have these characteristics :(i) low toxicity and serum-stable RNA loading methods, (ii) accurate targeting of target genes, and (iii) a high degree of uptake by target cells and endosomal escape. Through simple structural modification and packaging of the basic nanomaterial carrier, such as the use of specific cell membrane vesicles to make it targeted and biosafe, the carrier can achieve the multifunctional requirements containing the above characteristics. The optimization of nanomaterials will provide a safe and effective method of RNA delivery, which will make RNA have a broader application prospect in cancer treatment. Although there are still some unsolved problems, it is believed that with the deepening of research, the continuous improvement of material understanding, and the increasing exploitation of new materials, safe and efficient RNA delivery solutions will be applied to the clinic in the near future. It is also hope that RNA therapy will be more widely adopted as a new class of therapies in combination with other therapies or drugs. At present, the mechanism of RNA therapy is also constantly revealed and explored, which is expected to shine in cancer treatment.

Reviewer 4 Report

I think the review is comprehensive, well written and treats a subject that will be of interest for the readership of Bioengineering journal. For these reasons I believe the review deserves publication in the journal after some typos will be addressed.

Author Response

I think the review is comprehensive, well written and treats a subject that will be of interest for the readership of Bioengineering journal. For these reasons I believe the review deserves publication in the journal after some typos will be addressed.

Response: We thank the reviewers for their positive comments and recognition of our work and provide suggestions for further improving the quality of the manuscript. We have repeatedly revised and polished the language of the manuscript, hoping to effectively improve the quality of the manuscript.

Round 2

Reviewer 2 Report

The authors have addressed my concerns.

Reviewer 3 Report

According to the suggestions, the author supplements the background introduction about the origin and development of RNA as a gene drug. In the outlook section, author added the importance of improving the delivery strategy of nanomaterials as a carrier and suggestions for transformation. At the same time, grammatical and spelling errors in the article have been corrected. In summary, the paper has been greatly improved, with clearer logic and more complete expression, and I have no problem in recommending it for publication.